# Periodontal Disease Diagnosis in the Context of Oral Rehabilitation Approaches

**Laura Elisabeta Checherita** [1,†], **Magda Ecaterina Antohe** [2,*,†], **Ovidiu Stamatin** [2,*,†], **Ioana Rudnic** [1,*,†], **Iulian Costin Lupu** [2,†], **Irina Croitoru** [3,†], **Amelia Surdu** [2,†], **Daniel Cioloca** [4,†], **Irina Gradinaru** [2,†], **Laurian Francu** [4,†], **Iolanda Foia** [2,†], **Bogdan Mihai Vascu** [2,†] and **Ana Maria Fătu** [2,†]

[1]  2nd Dental Medicine Department, Faculty of Dental Medicine, "Grigore T. Popa" University of Medicine and Pharmacy, 700115 Iasi, Romania

[2]  3rd Dental Medicine Department, Faculty of Dental Medicine, "Grigore T. Popa" University of Medicine and Pharmacy, 700115 Iasi, Romania

[3]  Department of Foreign Languages, Faculty of Dental Medicine, "Grigore T. Popa" University of Medicine and Pharmacy, 16 Universității Street, 700115 Iasi, Romania

[4]  1st Dental Medicine Department, Faculty of Dental Medicine, "Grigore T. Popa" University of Medicine and Pharmacy, 700115 Iasi, Romania

*  Correspondence: magda.antohe@umfiasi.ro (M.E.A.); ovidiustamatin@yahoo.com (O.S.); ioana.rudnic@umfiasi.ro (I.R.); Tel.: +40-742843465 (M.E.A.); +40-744192605 (O.S.); +40-758662019 (I.R.)

†  These authors contributed equally to this work.

**Abstract:** Periodontal diseases generally correspond to a disturbance in the balance between the host's defense and the micro-organisms colonizing the periodontal environment. The exact mechanisms underlying the destruction of the periodontium remain to be fully elucidated. Our study aims to quantify the main bacteria pool involved in periodontal pathology and associate it with other factors involved in the onset of periodontal disease so that an accurate diagnosis with profound implications for the therapeutic algorithm can be developed. Micro-Ident tests, based on the polymerase chain reaction (PCR) technique, were used for the study group, chosen for their high specificity in identifying periodontopathogenic bacteria and determining their relative numbers. The results of our study indicate an increased concentration of 4.50 (number of strains) for Capnocytophaga, followed by Tannerella forsythia, in a concentration of 3.50; the next highest concentration percentages are for Treponemei denticola, and Prevotela intermedia, low concentrations were found for Fusobacterium nucleatum and Porphyromonas. The concentration of each type of bacteria is reflected in the clinical picture and constitutes the starting point for a targeted antibiotic therapy. Following the effects of antibiotic-targeted therapy obtained from the evaluation of the micro-IDent B test results on the periodontium of the supporting teeth, we observed that the values of the periodontal indices change slightly at 3-month intervals with a predominance of plaque, bleeding, and gingival indices, and less in the indices concerning the depth of the probing pocket and the loss of attachment on the buccal and oral surfaces. In conclusion, our study emphasizes a direct relationship between the subgingival tartar presence and the patients age, gingival recession, presence of periodontal pockets, dental mobility, as well as the periodontal indexes: plaque index, bleeding index, and gingival index. The correlation of negative values of periodontal indices with the nature of the involved bacteria materializes in relevant starting points in the elaboration of the periodontal diagnosis of the therapeutic plan and predictability of the prognosis of oral rehabilitation.

**Keywords:** micro-Ident tests; qPCR test; therapy; stomatognathic system pathology; oral rehabilitation; diagnosis; periodontal diseases

## 1. Introduction

Oral rehabilitation, a particularly complex integrative concept of contemporary dental practice, is based on new therapeutic approaches, in which the etiopathogenic factor is

of particular significance in establishing a precise diagnosis as the starting point for an individualized treatment plan [1]. The degree of periodontal damage decisively influences the predictability of rehabilitation therapy in a multifactorial context [2]. Considered in the past as "essential" diseases, inflammatory diseases of the marginal and deep periodontium nowadays recognize a specific microbial etiology, with bacterial plaque as the primary factor, to which a collection of favoring and systemic factors are added [3]. Periodontal diseases generally correspond to a disturbance in the balance between the host's defense and the micro-organisms colonizing the periodontal environment [4]. The exact mechanisms by which destruction of the periodontium occurs remain to be fully elucidated. A multitude of etiological factors leads to the inclusion of periodontitis in the group of complex multifunctional etiology [5]. Periodontal disease is a mixed infection, caused by the microbiota of the bacterial plaque and its toxic products, and in this respect, it should be noted that there is not a single bacterium or a single product responsible for tissue destruction, but several, acting either synergistically or at different stages of the evolution of periodontal disease [6]. Some bacterial species are not directly responsible for the onset of the infectious process, but they maintain it and generate optimal conditions for promoting the virulent potential of others [7,8]. The host organism's response to the action of the microbial factor is commensurate with its complexity and results in inflammation and infection, which are designed to protect the cells [9,10]. The extent of the host's immune response, however, also affects the body's own structures, leading in part to their destruction and the progression of periodontal lesions [11]. The particularities of inflammation at this level are also related to the anatomy and physiology of the periodontium [12,13]. Recognition of bacterial plaque site differences in various clinical patterns (disease vs. health) led to a renewed search for specific pathogens in periodontal disease and a conceptual transition from the nonspecific to the specific plaque hypothesis. [14,15] The source of periodontal risk factors was discussed in detail in recent years. Risk factors may influence a subject in general or may affect localized periodontal tissues [16]. Individual variability in periodontal tissue destruction and documented longitudinally in untreated populations required certain procedures for diagnosis in order to identify early subjects at increased risk for severe periodontal disease [17,18]. Monitoring untreated disease by investigating periodontal pocket depth, clinical attachment level, or bleeding on probing has limited value for indicating present activity or predicting future attachment loss [19]. On the other hand, the real impact of short bursts of activity on cumulative periodontal tissue loss also remains to be determined [20,21]. Continued mild attachment loss may have considerable consequences eventually, although it is un-detectable in studies limited to a duration of a few months [22]. To give accuracy of periodontal probing, detection of a continuous disease process, leading to a 6 mm attachment loss over 60 years, may require a minimum study period of 20 years [23]. From the range of usual methods of quantifying periodontal damage in dental practice, subjective observational indices have an important significance, such as Loe's and Silness' indices, which are based on certain criteria: bleeding on light probing, pocket depth on probing, loss of clinical attachment level, and radiographic evidence of alveolar bone loss [24]. Unfortunately, these clinical indicators, with the exception of bleeding on probing, are generally unable to reflect past disease and previous lesions, and cannot measure disease activity [25,26]. Bleeding of gingival tissue on probing at the sulcular or pocket gingival margin is considered to be a primary indicator correlating with active periodontal disease [27,28]. There are, however, concerns that bleeding is itself a non-subjective criterion of disease and its diagnostic value is questioned, being associated with increased percentages of false positive indicators of periodontal disease [29,30].

Periodontal disease is a particularly complex condition with profound systemic implications, both in terms of the inflammatory mediators it produces, and the microbial elements involved, which can enter the systemic circulation from the oral cavity and can affect various organs at a distance. The literature provides concrete links between periodontal disease and various common systemic pathologies, such as diabetes, cardiovascular diseases, respiratory diseases, obesity, neurological diseases, neoplastic pathologies, etc. [31,32]. From

the range of the influences of the general status on the dynamics of periodontal pathology, the effects of low vitamin D levels on periodontal health are notable. In recent decades, it was found that vitamin D deficiency is associated with varying degrees of severity of periodontal disease. A number of studies demonstrated that the beneficial effects of vitamin D can be found in the reduction in Porphyromonas gingivalis by active autophagy [33,34].

## 2. Aim of the Study

Our study aims to quantify the main bacteria involved in periodontal pathology and associate them with the other factors involved in the onset of periodontal disease so that an accurate diagnosis can be developed, with profound implications for the therapeutic plan.

The recording of gingival bleeding and plaque index, both at the beginning of treatment and along the way, is an objective tool for assessing the success of periodontal therapy. Additionally, it creates the possibility of prosthetic restoring means of arch integrity to prevent the occurrence of edentulousness and their complications is envisaged. Our findings emphasize the great importance of disease prevention and oral/periodontal healthcare for general wellbeing during a lifetime.

## 3. Materials and Methods

The study design was built according to the methodology of case control studies, where the selection of patients was consecutive for those presented in person for oral rehabilitation therapy in accordance with the Ethics Committee of "Grigore T. Popa" University of Medicine and Pharmacy Iasi. The sample size of 390 patients was established, taking into consideration the size of population in Iasi County (around 900.000 inhabitants) and the prevalence of periodontal disease in Romania, which is around 40–50%, for a confidence level of 95% and a population proportion of 50% (standard value).

Inclusion criteria for the cases were: patients suffering from periodontal disease in different degrees, where rehabilitation treatments were carried out.

The exclusion criteria for the cases were: non-periodontal disease; non-cooperating patients; advanced or terminal illnesses; post radiation oncology with poor blood coagulability grades; incapacitated for scheduled procedures; those with medication that induce periodontal damage; and also patients under antibiotic therapy supportive general treatments for at least 3 months. The control group united patients who never underwent targeted antibiotic therapy dictated by the bacteria involved in the onset of periodontal pathology.

### 3.1. Paraclinical Section

Micro-IDent tests, based on the polymerase chain reaction (PCR) technique, were used for the study group, chosen for their high specificity in identifying periodontopathogenic bacteria and determining their relative numbers. The mentioned tests were chosen over bacterial cultures as they are able to use DNA-like germ identification, irrespective of their viability (anaerobic bacteria often resist very little during transport to the laboratory and cultures can give false negative results for periodontal pathogens).

The micro-IDent test made it possible to determine the five important periodontal organisms: *Actinobacillus actinomycetemcomitans*, *Porphyromonas gingivalis*, *Prevotella intermedia*, *Bacteroides forsythus*, and *Treponema denticola*, and their correlation with the depth of the periodontal pocket, which is the basis for a precise diagnosis and therapy. In some cases of periodontitis, the diagnostic spectrum could be broadened by determining six other germs: *Peptostreptococcus micros*, *Fusobacterium nucleatum/periodonticum*, *Eikenella corrodens*, *Campylobacter rectus*, *Eubacterium nodatum*, and *Capnocytophaga* spp. (micro-IDent B test).

### 3.2. Clinical Section

The methodology of clinical evaluation implies the index quantification as follows:

The clinical periodontal status of the patients included in the study was established visually and by probing through the recording the following parameters at selected tooth sites: gingival index (GI Löe and Silness), plaque index (PI), bleeding on sulcular probing

(BI) pocket depth on probing (PDP), and level of attachment loss (CAL), measured with a periodontal probe graduated from the enamel–cement junction to the most apical penetration of manual probing. Bone resorption was determined on the basis of clinical and radiographic criteria, which provided the necessary information to include or exclude cases from the study. Patients were clinically and periodontally reassessed at 3 and 6 months after application of specific therapy in accordance with the results of micro-IDent tests.

GCF (gingival crevicular fluid) collection was performed using sterile, absorbent, filter paper strips of standardized 2 × 8 mm size with slightly rounded tips inserted into the gingival sulcus in the buccal and oral sites of the abutment/anchor teeth bordering the edentulousness (Figure 1).

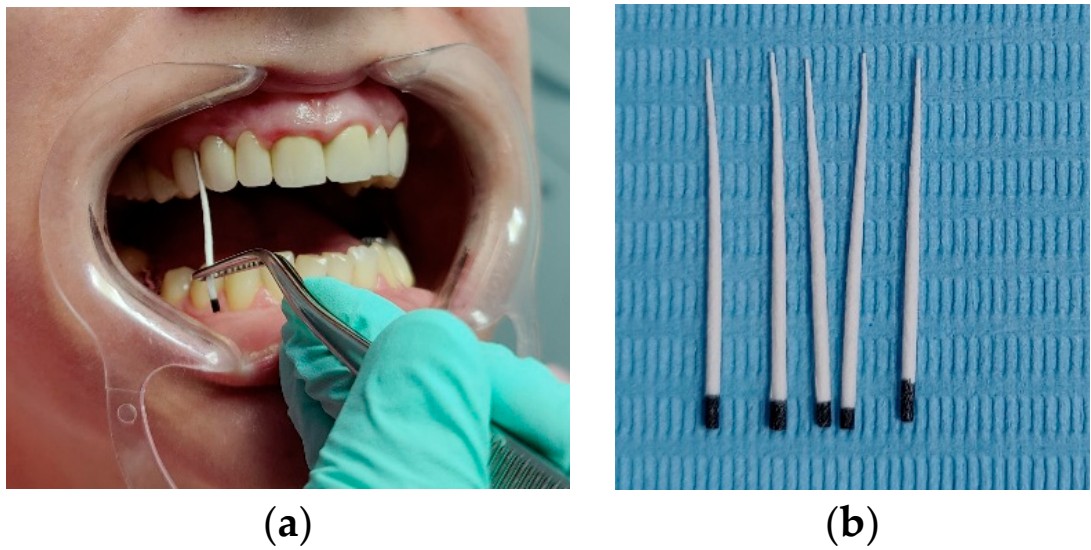

**(a)** **(b)**

**Figure 1.** Practical aspects of GCF sampling. (**a**) the action of performing GCF sampling collection in premolar area in first quadran maxilla; (**b**) sample of filter paper strips of standardized 2 × 8 mm size with slightly rounded tips used in the performing GCF sampling action.

*3.3. Statistical Analysis*

The statistical analysis was performed in Statistica 14.0. The Kolmogorov–Smirnov test was used to check the distribution of the numerical variables. The numerical variables were expressed through average values and standard deviations, and the categorial values were expressed through absolute frequencies and percentages. The Pearson chi-squared test was used to compare categorial variables within samples, and Mann–Whitney and Wilcoxon Signed Rank tests were used to compare numerical variables with non-normal distributions within samples. The value $p < 0.05$ was considered statistically significant and the value $p < 0.01$ was considered highly statistically significant. The Pearson R correlation coefficient was used to investigate the relation between numerical variables.

## 4. Results

Clinical and laboratory evaluation was conducted between 2018 and 2022 on a group of 250 women and 140 men. Subjects were aged between 20 and 70 years, from "Mihail Kogalniceanu" Clinical Base of Dentistry Education Iasi, "Grigore T. Popa" University of Medicine and Pharmacy Iasi, and also "Apollonia" University from Iasi, of which 35 subjects following periodontal therapy in the context of oral complex rehabilitation treatments were selected. The group of patients was divided into a study group and a control group, each group consisting of 125 women and 70 men; the patients in control group did not receive antibiotics for the periodontal pathology, and the patients in the study group were divided in two other sub-samples: patients who received nonspecific antibiotics and a sub-sample of 35 patients who received specific antibiotics according to the bacteria involved in the onset of periodontal pathology.

The general characteristics of the samples are presented in Table 1.

**Table 1.** Demographic parameters of the sample.

| Parameter | Total (n = 390) | | Study Sample (n = 195) | | Control Sample (n = 195) | | *p*-Value |
|---|---|---|---|---|---|---|---|
| | n | % | n | % | n | % | |
| Gender | | | | | | | 1.000 † |
| M | 140 | 3.9 | 70 | 35.9% | 70 | 35.9% | |
| F | 250 | 64.1 | 125 | 64.1% | 125 | 64.1% | |
| Age group | | | | | | | 0.919 † |
| 20–29 yrs | 43 | 11.0% | 21 | 10.8% | 22 | 11.3% | |
| 30–39 yrs | 45 | 11.5% | 20 | 10.3% | 25 | 12.8% | |
| 40–49 yrs | 63 | 16.2% | 33 | 16.9% | 30 | 15.4% | |
| 50–59 yrs | 94 | 2.1% | 46 | 23.6% | 48 | 24.6% | |
| 60–69 yrs | 145 | 37.2% | 75 | 38.5% | 70 | 35.9% | |
| Subgingival tartar | | | | | | | 0.919 † |
| absent | 221 | 56.7% | 110 | 56.4% | 111 | 56.9% | |
| present | 169 | 43.3% | 85 | 43.6% | 84 | 43.1% | |
| Gingival recession | | | | | | | 0.463 † |
| absent | 245 | 62.8% | 119 | 61.0% | 126 | 64.6% | |
| present | 145 | 37.2% | 76 | 39.0% | 69 | 35.4% | |
| Periodontal pockets | | | | | | | 0.059 † |
| absent | 144 | 36.9% | 81 | 41.5% | 63 | 32.3% | |
| present | 246 | 63.1% | 114 | 58.5% | 132 | 67.7% | |
| Dental mobility | | | | | | | <0.001 **,† |
| 0 | 179 | 45.9% | 106 | 54.4% | 73 | 37.4% | |
| 1 | 123 | 31.5% | 63 | 32.3% | 60 | 30.8% | |
| 2 | 66 | 16.9% | 23 | 11.8% | 43 | 22.1% | |
| 3 | 22 | 5.6% | 3 | 1.5% | 19 | 9.7% | |
| Plaque index PI (m ± SD) | 0.354 ± 0.252 | | 0.376 ± 0.248 | | 0.333 ± 0.255 | | 0.085 ‡ |
| Bleeding index BI (m ± SD) | 0.361 ± 0.247 | | 0.479 ± 0.228 | | 0.244 ± 0.206 | | <0.001 **,‡ |
| Gingival index GI (m ± SD) | 0.370 ± 0.243 | | 0.466 ± 0.208 | | 0.275 ± 0.239 | | <0.001 **,‡ |
| PDP index (m ± SD) | 2.630 ± 0.191 | | 2.522 ± 0.212 | | 2.722 ± 0.102 | | <0.001 **,‡ |

† Pearson chi-squared test; ‡ Mann–Whitney test; ** $p < 0.01$ highly statistically significant.

The gender structure of the two samples analysed is identical—the female gender prevails; in both samples, adult and elderly patients over 50 years of age also prevail (62.1% in the study group and 60.5% in the control group). Subgingival tartar was found in equal relative proportions in both groups (ca. 40%), as were gingival recessions and periodontal pockets, observed in ca. 60% of cases in both groups. The majority of the patients present dental mobility grade 1 or 2 (48.4%). As regards the periodontal evaluation indices investigated, the plaque index is slightly higher in the study group compared to the control group, as well as the bleeding index and the gingival index—in the last two cases the difference was even statistically significant. The depth of the periodontal pockets is, on the other hand, greater in the control group compared to the study group.

The initial clinical examination included the registration of plaque and bleeding index, as well as other changes in the gums: color change (intense red, red–purple), change in gum contours, gingival texture, and consistency. The position of the gums, the existence of recessions and bags were also monitored.

The assessment of plaque deposits was made using the Quigley–Hein colored plaque index, and to determine the gingival inflammation located at the level of the papillae, we used the papillary bleeding index—PBI of Saxen and Muhlemann.

In this session, we proceeded to highlight the bacterial plaque using the following solutions as revelers:

- 2% methyl blue,
- 1% fuchsia solution,
- fluorescent substances (Figure 2)

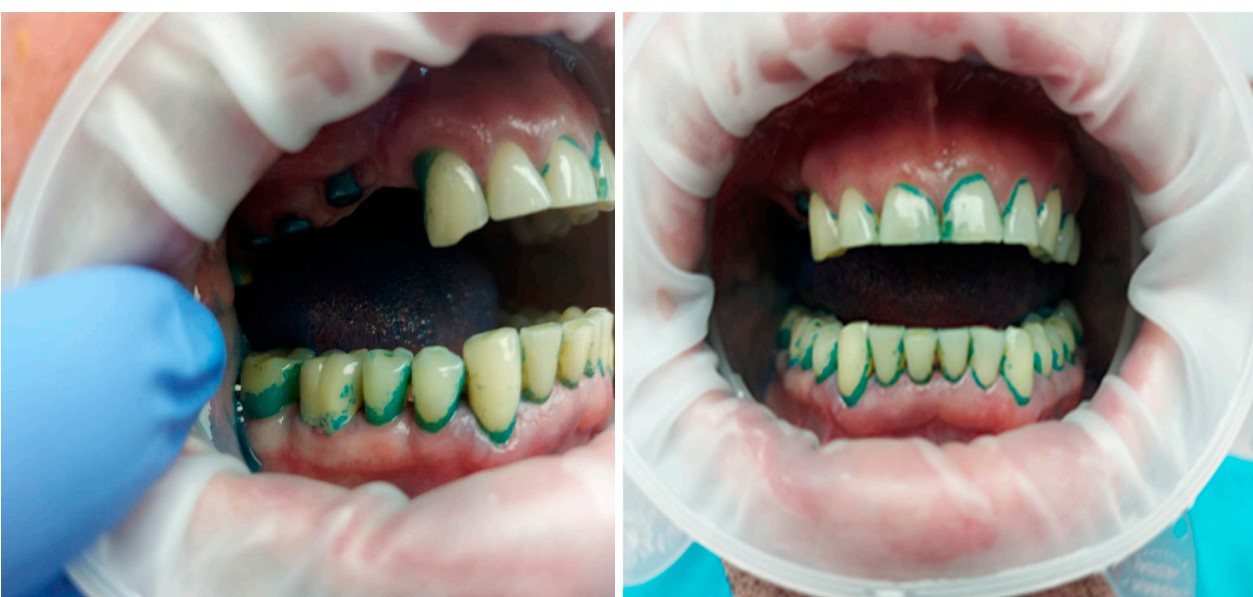

**Figure 2.** Patient D.B-methyl blue solution.

For evaluation, we correlated the prevalence of subgingival tartar deposits with the clinical symptoms that are characteristic of periodontal disease and with the demographic parameters of the samples, at the level of the entire patient lot and comparative in the study and control groups (Table 2). No statistically significant differences were observed between genders in the presence of subgingival calculus. The study by age group shows statistically significant differences, both at the level of the whole group and separately by subgroups, with patients with tartar belonging mainly to the age group over 60 years. Furthermore, patients with tartar also showed significantly increased percentages of gingival recession (both overall, 62.1%, and separated by sublots, 62.4%, and 61.9%, respectively), periodontal pockets (overall, 82.2% and separated by sublots, 87.1%, and 77.4%, respectively), and tooth mobility (grade 1 in the study group, 52.9%, and grade 1, 26.2%, and grade 2, 28.6%, in the control group). The plaque index was significantly higher overall in patients with tartar compared to the others ($0.416 \pm 0.293$ compared to $0.307 \pm 0.204$), a difference that was also found separately in the two samples but was more attenuated in the study group ($0.386 \pm 0.253$ compared to $0.368 \pm 0.244$) and accentuated in the control group ($0.446 \pm 0.328$ compared to $0.248 \pm 0.130$). A similar behaviour was observed for the bleeding index and the gingival index. No large discrepancies were observed in the depth of periodontal pockets in patients with tartar compared to the others at the level of the whole group analysed ($2.623 \pm 0.201$ compared to $2.638 \pm 0.177$), and in particular, in the study group ($2.520 \pm 0.212$ compared to $2.529 \pm 0.226$); in contrast, in the control group, the depth of periodontal pockets is greater in patients with tartar ($2.741 \pm 0.110$) than in the others ($2.704 \pm 0.091$).

Our study focused on the comparative evaluation of the outcome of patients diagnosed with periodontal disease over a 6-month period, with interim evaluation at 3 months, in the absence of antibiotic therapy (control group) versus antibiotic administration (study group). Literature data highlight that specific antibiotic therapy, depending on the prevalent bacteria, leads to optimal long-term outcomes, and a current therapeutic trajectory is the working hypothesis for our study. In order to validate this hypothesis, we quantified the results of the micro-IDent analysis by counting the strains of bacteria involved in the 35 patients selected from the study group; the results obtained are shown in Table 3 and Figure 3. One can notice an increased concentration of 4.50 (number of strains) for Capnocytophaga, followed by Tannerella forsythia, in a concentration of 3.50. The next highest concentration percentages are for Treponemei denticola and Previtelei intermedia; low concentrations were found for Fusobacterium nucleatum and Porphyromonas; and the

concentration of each type of bacteria is reflected in the clinical picture and constitutes the starting point for the targeted antibiotic therapy.

**Table 2.** The relation between subgingival tartar presence and the other investigated parameters—univariate analysis.

| Parameter | Total (n = 390) | | Subgingival Tartar Study Sample (n = 195) | | Control Sample (n = 195) | |
|---|---|---|---|---|---|---|
| | Absent n (%) | Present n (%) | Absent n (%) | Present n (%) | Absent n (%) | Present n (%) |
| Gender | $p = 0.776$ [†] | | $p = 0.294$ [†] | | $p = 0.516$ [†] | |
| M | 78 (35.3%) | 62 (36.7%) | 36 (32.7%) | 34 (40.0%) | 42 (37.8%) | 28 (33.3%) |
| F | 143 (64.7%) | 107 (63.3%) | 74 (67.3%) | 51 (60.0%) | 69 (62.2%) | 56 (66.7%) |
| Age group | $p < 0.001$ [**,†] | | $p = 0.003$ [**,†] | | $p = 0.003$ [**,†] | |
| 20–29 yrs | 41 (18.6%) | 2 (1.2%) | 20 (18.2%) | 1(1.2%) | 21 (18.9%) | 1 (1.2%) |
| 30–39 yrs | 27 (12.2%) | 18 (10.7%) | 12 (10.9%) | 8 (9.4%) | 15 (13.5%) | 10 (11.9%) |
| 40–49 yrs | 36 (16.3%) | 27 (16.0%) | 19 (17.3%) | 14 (16.5%) | 17 (15.3%) | 13 (15.5%) |
| 50–59 yrs | 47 (21.3%) | 47 (27.8%) | 23 (20.9%) | 23 (27.1%) | 24 (21.6%) | 24 (28.6%) |
| 60–69 yrs | 70 (31.7%) | 75 (44.4%) | 36 (32.7%) | 39 (45.9%) | 34 (30.6%) | 36 (42.9%) |
| Gingival recession | $p < 0.001$ [**,†] | | $p < 0.001$ [**,†] | | $p < 0.001$ [**,†] | |
| absent | 181 (81.9%) | 64 (37.9%) | 87 (79.1%) | 32 (37.6%) | 94 (84.7%) | 32 (38.1%) |
| present | 40 (18.1%) | 105 (62.1%) | 23 (20.9%) | 53 (62.4%) | 17 (15.3%) | 52 (61.9%) |
| Periodontal pockets | $p < 0.001$ [**,†] | | $p < 0.001$ [**,†] | | $p = 0.012$ [*,†] | |
| absent | 114 (51.6%) | 30 (17.8%) | 70 (63.6%) | 11 (12.9%) | 44 (39.6%) | 19 (22.6%) |
| present | 107 (48.4%) | 139 (82.2%) | 40 (36.4%) | 74 (87.1%) | 67 (60.4%) | 65 (77.4%) |
| Dental mobility | $p < 0.001$ [**,†] | | $p < 0.001$ [**,†] | | $p < 0.001$ [**,†] | |
| 0 | 140 (63.3%) | 39 (23.1%) | 86 (78.2%) | 20 (23.5%) | 54 (48.6%) | 19 (22.6%) |
| 1 | 56 (25.3%) | 67 (39.6%) | 18 (16.4%) | 45 (52.9%) | 38 (34.2%) | 22 (26.2%) |
| 2 | 24 (10.9%) | 42 (24.9%) | 5 (4.5%) | 18 (21.2%) | 19 (17.1%) | 24 (28.6%) |
| 3 | 1 (0.5%) | 21 (12.4%) | 1 (0.9%) | 2 (2.4%) | 0 (0.0%) | 19 (22.6%) |
| Plaque index PI (m ± SD) | $p = 0.001$ [**,‡] | | $p = 0.602$ [‡] | | $p < 0.001$ [**,‡] | |
| | 0.307 ± 0.204 | 0.416 ± 0.293 | 0.368 ±0.244 | 0.386 ± 0.253 | 0.248 ± 0.130 | 0.446 ± 0.328 |
| Bleeding index BI (m ± SD) | $p = 0.001$ [**,‡] | | $p = 0.328$ [‡] | | $p < 0.001$ [**,‡] | |
| | 0.323 ± 0.230 | 0.411 ± 0.260 | 0.464 ± 0.225 | 0.497 ± 0.232 | 0.184 ± 0.128 | 0.324 ± 0.258 |
| Gingival index GI (m ± SD) | $p < 0.001$ [**,‡] | | $p = 0.207$ [‡] | | $p < 0.001$ [**,‡] | |
| | 0.330 ± 0.234 | 0.422 ± 0.246 | 0.452 ± 0.208 | 0.484 ± 0.207 | 0.210 ± 0.193 | 0.360 ± 0.267 |
| PDP index (m ± SD) | $p = 0.754$ [‡] | | $p = 0.766$ [‡] | | $p = 00.029$ [*,‡] | |
| | 2.638 ± 0.177 | 2.623 ± 0.201 | 2.529 ± 0.226 | 2.520 ± 0.212 | 2.704± 0.091 | 2. 741 ± 0.110 |

[†] Pearson chi-squared test; [‡] Mann–Whitney test; [*] $p < 0.05$ statistically significant; [**] $p < 0.01$ highly statistically significant.

**Table 3.** Bacteria identified in the study sample.

| | Study Sample—With Bacteria Counting (n = 35) | |
|---|---|---|
| | n (%) | m ± SD |
| *Tannerella forsythia* | 27 (77.1%) | 3.554 ± 0.946 |
| *Treponema denticola* | 32 (91.4%) | 2.883 ± 0.782 |
| *Prevotella intermedia* | 22 (62.9%) | 2.763 ± 0.865 |
| *Fusobacterium nucleatum* | 30 (85.7%) | 1.328 ± 0.439 |
| *Capnocytophaga* | 12 (34.3%) | 4.547 ± 1.068 |
| *Porphyromonas* | 28 (80.0%) | 1.658 ± 0.554 |

The microbiota associated with periodontal health and disease was studied with a wide variety of sampling techniques and bacterial culture, as well as different classifications of disease status.

This bacterial prevalence, together with the histopathological analysis of the glandular acini, provides the basis for a diagnosis of choice in cases requiring oral rehabilitation (Figures 4 and 5).

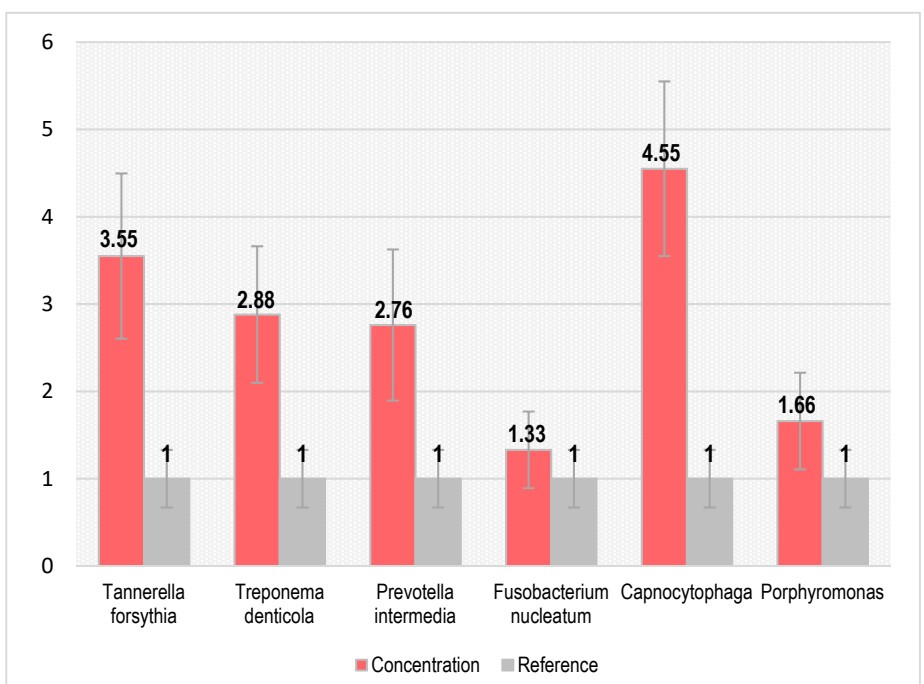

**Figure 3.** Concentration of bacteria type involved in periodontal pathology of study group.

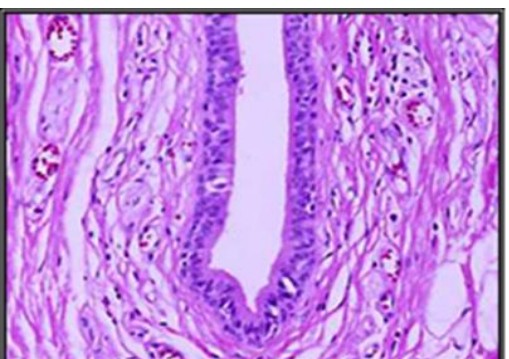

**Figure 4.** Microscopic appearance of an excretory duct from the submandibular gland in adults (H&E) with periodontal diseases.

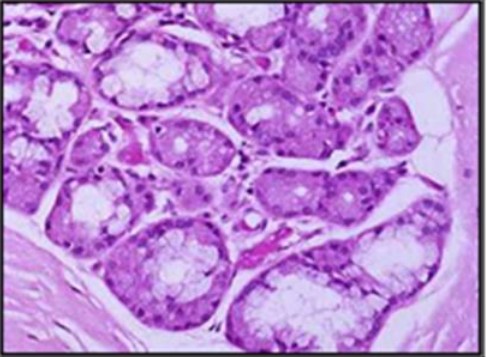

**Figure 5.** Presentation of a submandibular gland in the elderly with marked berries, inequality dimensionally reduced serous berries, with periodontal disease.

In the control group, without antibiotic therapy, a significant worsening of periodontal disease was observed, as expected, both at the 3-month and the 6-month evaluation, supported by an increase in the values of the four periodontal indices assessed (Table 4). In

the patients of the study group who received non-specific antibiotic therapy, there was a significant decrease in the values of the periodontal indices, with two observations: the gingival index decreased significantly at the 3-month evaluation compared to the initial time, but still had a constant value level without significant variation, and the PDP index, after a significant decrease at 3 months, worsened slightly at 6 months without reaching the initial level—an obviously positive fact. In the patients of the study group who received specific antibiotic therapy, the plaque index decreased significantly compared to the initial time both at the 3-month and at the 6-month evaluation, the bleeding index decreased significantly at the 3-month evaluation, after which it remained relatively constant, the gingival index showed a significant decrease at the 3-month assessment, followed by a slight worsening at the 6-month assessment, but remained significantly lower than at the initial time, and the PDP index decreased steadily at both monitoring times.

**Table 4.** The comparative evolution of periodontal indexes in the investigated samples.

| Index (m ± SD) | Time | | | *p*-Value [†] | | |
| --- | --- | --- | --- | --- | --- | --- |
| | Initially | 3 Months | 6 Months | 0 vs. 3 m | 3 m vs. 6 m | 0 vs. 6 m |
| Control sample (n = 195) | | | | | | |
| Plaque index PI | 0.333 ± 0255 | 0.473 ± 0.504 | 0.685 ± 0.668 | $p < 0.001$ **,[†] | $p < 0.001$ **,[†] | $p < 0.001$ **,[†] |
| Bleeding index BI | 0.244 ± 0206 | 0.530 ± 0.101 | 0.550 ± 0.100 | $p < 0.001$ **,[†] | $p < 0.001$ **,[†] | $p < 0.001$ **,[†] |
| Gingival index GI | 0.275 ± 0239 | 0.415 ± 0.200 | 0.522 ± 0.317 | $p < 0.001$ **,[†] | $p < 0.001$ **,[†] | $p < 0.001$ **,[†] |
| PDP index | 2.722 ± 0102 | 3.210 ± 0.201 | 3.640 ± 0.803 | $p < 0.001$ **,[†] | $p < 0.001$ **,[†] | $p < 0.001$ **,[†] |
| Study sample—without bacteria counting (n = 160) | | | | | | |
| Plaque index PI | 0.381 ± 0247 | 0.360 ± 0.206 | 0.354 ± 0.202 | $p < 0.001$ **,[†] | $p < 0.001$ **,[†] | $p < 0.001$ **,[†] |
| Bleeding index BI | 0.481 ± 0219 | 0.471 ± 0.260 | 0.463 ± 0.247 | $p = 0.006$ ** | $p < 0.001$ **,[†] | $p < 0.001$ **,[†] |
| Gingival index GI | 0.465 ± 0195 | 0.433 ± 0.203 | 0.434 ± 0.220 | $p < 0.001$ **,[†] | $p = 0.868$ | $p < 0.001$ **,[†] |
| PDP index | 2.538 ± 0179 | 2.382 ± 0.260 | 2.420 ± 0.187 | $p < 0.001$ **,[†] | $p = 0.004$ ** | $p < 0.001$ **,[†] |
| Study sample—with bacteria counting (n = 35) | | | | | | |
| Plaque index PI | 0.352 ± 0252 | 0.334 ± 0.214 | 0.327 ± 0.214 | $p = 0.015$ * | $p = 0.005$ ** | $p = 0.006$ ** |
| Bleeding index BI | 0.467 ± 0269 | 0.454 ± 0.306 | 0.451 ± 0.288 | $p = 0.097$ | $p = 0.853$ | $p = 0.005$ ** |
| Gingival index GI | 0.467 ± 0260 | 0.440 ± 0.270 | 0.450 ± 0.284 | $p < 0.001$ **,[†] | $p = 0.025$ * | $p = 0.021$ * |
| PDP index | 2.489 ± 0273 | 2.359 ± 0.334 | 2.269 ± 0.375 | $p < 0.001$ **,[†] | $p < 0.001$ **,[†] | $p < 0.001$ **,[†] |

[†] Wilcoxon Signed Ranks test; * $p < 0.05$ statistically significant; ** $p < 0.01$ highly statistically significant.

Following the effects of antibiotic-targeted therapy obtained from the evaluation of the micro-IDent B test results on the periodontium of the supporting teeth, we observed that the values of the periodontal indices change slightly at 3-month intervals with a predominance of plaque, bleeding, and gingival indices, and less in the indices concerning the depth of the probing pocket and the loss of attachment on the buccal and oral surfaces. Immobilization of affected teeth with polyethylene fibre and composite increases their stability and slows down the rate of resorption.

In order to quantify the impact of bacterial identification in the therapeutic plan, we also investigated the degree of correlation of bacterial strains with the evolution of periodontal indices (Table 5).

**Table 5.** Correlation between the bacteria amount and the periodontal disease.

| | Plaque Index PI | | Bleeding Index BI | | Gingival Index GI | | PDP Index | |
| --- | --- | --- | --- | --- | --- | --- | --- | --- |
| | r | *p* | r | *p* | r | *p* | r | *p* |
| *Tannerella forsythia* | 0.354 | 0.070 | 0.274 | 0.167 | 0.126 | 0.532 | 0.194 | 0.332 |
| *Treponema denticola* | 0.393 | 0.026 * | 0.322 | 0.072 | 0.329 | 0.066 | 0.381 | 0.031 * |
| *Prevotella intermedia* | 0.388 | 0.075 | 0.427 | 0.048 * | 0.465 | 0.029 * | 0.446 | 0.038 * |
| *Fusobacterium nucleatum* | 0.362 | 0.049 * | 0.277 | 0.138 | 0.130 | 0.495 | 0.081 | 0.670 |
| *Capnocytophaga* | 0.776 | 0.003 * | 0.662 | 0.019 * | 0.605 | 0.037 * | 0.431 | 0.162 |
| *Porphyromonas* | 0.381 | 0.045 * | 0.335 | 0.081 | 0.293 | 0.131 | 0.254 | 0.191 |

* $p < 0.05$ statistically significant.

Our study revealed directly proportional correlations between bacterial concentration and periodontal index values, which become statistically significant in some situations. Thus, increased concentrations of Treponema denticola are moderately but significantly

associated with worsening of the plaque index and PDP index, increased concentrations of Prevotella intermedia are moderately and significantly associated with worsening of the bleeding index, gingival index, and PDP index, increased concentrations of Fusobacterium nucleatum and Porphyromonas are moderately and significantly associated with worsening of the plaque index, and increased concentrations of Capnocytophaga are strongly and significantly associated with worsening of the plaque, bleeding, and gingival indices.

## 5. Discussion

The prevalence of total and subgingival tartar deposition increases with the age of patients [35].

Tartar acts on the periodontium in two ways: 1. By mechanical action that causes and maintains irritation at the gingival level, manifested by gingival ulceration, the appearance of granulation tissue. Tartar deposits can accumulate in large quantities on the surfaces of teeth, constituting real "blocks" that come into contact with the gingival margin, which they continually irritate, causing it to retract towards the apical, denuding new areas of root cement that become covered over time with tartar. Subgingival tartar has a mechanical irritating action on the gingival wall of the pocket, especially during mastication. 2. By the action of microorganisms contained in the bacterial plaque upon its surface, being a continuous source of infection. Microorganisms act on the gums either through their toxins or through immuno-allergic mechanisms. Enzymes released by bacteria are able to break the adhesion of epithelial cells, to lyse collagen, and to destroy the connective tissue matrix. Tartar provides a fixed site for the continuous accumulation of plaque that keeps it in contact with the gingiva [36].

The evaluation of the prevalence and the degree of loss of periodontal attachment (gingival recession and depth of periodontal pockets) demonstrates that there is a direct relationship with dental mobility [37].

As the gingival sulcus deepens subsequent to inflammatory phenomena, a new environment is created for colonization. In contrast to exposed tooth surfaces, the sulcus is protected from friction so that attachment and adhesion are less important for the flora than in young supragingival plaque [38]. The subgingival flora is adapted to an extremely different environment, having to use different nutrient principles, survive a lower oxygen concentration, and fight against the host's defensive mechanisms. The initial colonizers are Gram-negative cocci, bacilli, and spirochaetes, and newly formed crevicular or subgingival plaque may become stable as early as 4 weeks after the clinical onset of gingivitis. Gingival plaque becomes more complex with time and is mainly composed of obligate anaerobes or facultative anaerobes, such as *Porphyromonas*, *Eikenella*, *Vibrio*, *Selenomonas*, and *Capnocytophaga*. These species do not produce an extracellular matrix and therefore develop in a less adherent plaque. The literature shows that gingival bleeding is associated with an increase in Actinomyces and Porphyromonas gingivalis and Prevotella intermedia species. It is also proven that plaque accumulation involves a sequential ecological progression of bacterial exchange. When a certain level of bacterial complexity is reached, an increase in anaerobic Gram (−) species is observed, which is associated with the development of inflammation of the gingiva progressing towards the periodontium. Additionally, it is well known that saliva can reflect oral and systemic health status, while providing clinically important information for the various aspects of periodontal disease progression, while salivary qualitative changes can have important diagnostic value by identifying their active disease sites [39].

Quantitative bacterial loading of samples was obtained by comparing the fluorescent signal intensity of each species tested with the qPCR fluorescent signal of known bacterial concentrations (standard curve for *Porphiromonas gingivalis*). By extrapolating on the standard curve of each tested species, the number of bacterial cells collected from periodontal pockets was estimated.

Salivary acinar status is directly related to salivary quality and quantity, which can influence the course of periodontal disease. Thus, the analysis of the structure of glandular

acini prevalent in the submandibular glands, which were histopathologically evaluated in our study, provides data about changes in them due to age progression on the one hand, and on the other hand, there may be obstructive damage present from different etiologies. We observed changes in these glands with increasing age: towards the age range of 60–69 years, changes were present through acinar dimensional inequality in patients with periodontal pathology, correlating in a narrowing of the excretory duct. Histological changes in the submandibular glands through ductal narrowing and serous changes are associated in these patients in the age range of 60–69 years, with prevalence of Tannerella forsythia and Prevotella intermedia bacteria, which may be supported by qualitative and quantitative salivary changes in these patients.

However, the comparisons highlight the general characters of the microbial population currently found in different clinical states and implicate a discrete group of bacteria that function as periodontal pathogens. Studies using appropriate microbiological procedures clearly demonstrated that the number and proportion of different subgingival bacterial groups varied in periodontal health when compared to disease state [40,41]. The total number of bacteria determined by microscopic counts per gram of plaque are twice as high in pathological sites as in healthy sites. Since more plaque is thought to be found in diseased sites, this suggests that the total bacterial load is much higher than in healthy sites. Differences between periodontal health and disease are also evident when bacterial morphotypes from healthy and diseased sites are examined. Fewer cocci and more motile bacilli and spirochaetes are found in diseased sites compared to healthy sites. Bacteria cultured from healthy periodontal sites consist predominantly of Gram (+) facultative bacilli and cocci (about 75%). The recovery of this group of microorganisms decreases proportionally in gingivitis—44%, and in periodontitis—10–13%. These decreases are accompanied by an increase in the proportion of Gram (−) bacilli, from 13% in healthy areas to 40% in gingivitis, 65% in localized juvenile periodontitis, and 74% in advanced periodontitis [42]. The number of microorganisms was found to be double in affected sites compared to healthy sites. The initial values of the indices examined in the study group do not indicate severe periodontal changes with tissue destruction and intense inflammatory phenomena that could suggest active phases of a possible periodontal disease, but indicate, in some cases at the initial stage, various changes in clinical parameters triggered by the presence of different types of bacteria, as well as the presence of different degrees of mobility.

Furthermore, the minimal changes in gingival sulcus deepening observed with periodontal probing and the reduced change in attachment loss caused by the stress on the supporting teeth seem to be compensated by the supportive homeostatic mechanisms in young adults [43].

In the case of elderly patients, compared to the initial values of the clinical indicators of periodontal health, there is a greater increase in the values of the clinical parameters supporting gingivitis development after the first three months of oral rehabilitation with removable prostheses with inflammatory phenomena initiated and triggered by local irritation and retained plaque, but with values that are maintained for the bleeding index at 3 months with a subsequent increase at 6 months; the expansion is determined by the difficulties of hygiene measurements in the retentive areas created between the tooth and the elements of the removable prosthesis, in addition to the irritations and microtraumatic prosthesis that require a longer time, and the deep periodontium maintaining the inflammatory aspect. Thus, from the initial situation of "periodontal health" or with minimal changes in clinical parameters, significant changes in evolution are reached, reaching the aspect of gingivitis, mild, medium, and severe periodontitis, expressed by increased values of gingival indices, depth of pockets and loss of attachment level, and increased depth of the pocket compared to the initial situation, which is more important in the area under functional stress. In the elderly patient, we established correlates between periodontal indices and salivary acini status. The status of periodontal indices correlated with the status of salivary acini gives us trajectories of the predictability of the evolution of periodontal

disease, which is particularly important in determining long-term results in complex oral rehabilitation. Current aspects of complex oral rehabilitation lead to a clinical case approach at the end of the therapy, achieving a balance in the dynamic function at the TMJ level, in context of the dysfunctional syndrome of stomatognathic system pathology, DSSS syndrome [44].

TMJ functionality is correlated in turn with the morphological rehabilitation of dental arches affected by the occlusal parameter analysis in the context of habitual occlusion versus ideal occlusion.

This study is not free from limitations: first, the cross-sectional study design precludes it from external validity; second, even though all the patients referred to the clinical evaluation were screened and potentially eligible for the study, the dataset included a small sample. In addition, the decision to use different periodontal screening indices to evaluate oral health status was taken to increase the feasibility of the study; however, we are aware that a complete periodontal chart (with full-mouth plaque and full-mouth bleeding values), as well as radiographic and microbiological analysis, would provide additional information for this study.

## 6. Conclusions

Our study emphasizes a direct relationship between the subgingival tartar presence and the patients age, gingival recession, presence of periodontal pockets, dental mobility, as well as the periodontal indexes: plaque index, bleeding index, and gingival index. The subgingival tartar is significantly correlated with the PDP index only in the control sample.

In patients in the study group who received non-specific antibiotic therapy, there is a significant decrease in the value of the periodontal indices, with slight variations in the gingival index and PDP index, and in patients in the study group who received specific antibiotic therapy, the plaque index decreases significantly from baseline, both at the 3-month evaluation and at the 6-month evaluation, as well as the bleeding index and the PDP index, while the gingival index shows a significant decrease at the 3-month evaluation, followed by a slight worsening at the 6-month evaluation, but remains significantly lower than at the initial time.

Our study revealed directly proportional correlations between bacterial concentration and periodontal index values, which become statistically significant in some situations. Of all the bacteria involved, the presence of *Capnocytophaga* leads to longer-term results, unlike other bacteria involving a higher degree of local complications.

ID microdental tests are an extremely useful tool in the accurate diagnosis of the periodontal damage stage, with multiple implications for the complex rehabilitation therapy of each individual clinical case.

**Author Contributions:** Conceptualization, L.E.C., M.E.A. and A.M.F.; methodology, O.S. and I.R.; software, I.C. and I.C.L. validation L.E.C. and A.M.F.; formal analysis, D.C. and I.F.; investigation A.S., O.S. and I.R.; resources, M.E.A., D.C. and L.F. data curation, A.M.F., I.C.L. and L.F. writing—original draft preparation, M.E.A., I.F., L.F. and L.E.C.; writing—review and editing, I.G., B.M.V. and O.S.; visualization. I.G., B.M.V., I.F. and A.S.; supervision I.C.L., D.C. and I.C.; project administration I.F., B.M.V., I.C. and A.S. All authors have read and agreed to the published version of the manuscript.

**Funding:** This research received no external funding.

**Informed Consent Statement:** Informed consent was obtained from all subjects-rehabilitation treatments involved in the study.

**Data Availability Statement:** Not applicable.

**Conflicts of Interest:** The authors declare no conflict of interest.

**Abbreviations**

| | |
|---|---|
| PD | periodontal disease |
| SS | stomatognathic system |
| TMJ | temporomandibular joint |
| PCR | polymerase chain reaction |
| DSSS | dysfunctional syndrome of stomatognathic system |
| PET | pozitron emission test |
| GCF | gingival crevicularfFluid |
| qPCR | quantitative PCR (qPCR) test is used to detect, characterize, and quantify nucleic acids |

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
