# Peer review of "Periodontal Disease Diagnosis in the Context of Oral Rehabilitation Approaches"

_applsci, doi:10.3390/app12189067_

Round 1

Reviewer 1 Report

The reviewer really appreciates the efforts of the authors to conduct this study which has a good clinical significance. However, there are lots of scopes to improve the quality of the manuscript. The reviewer would like to suggest the following revision in the manuscript to make it suitable for publication.

·         The abstract needs to be re-structured by shortening the methodology and adding highlighted results and a clear conclusion/ outcome of the study.

·         The Authors have cited 3-6 articles for a single statement. The reviewer’s suggestion would be to use 1 or maximum 2 articles that precisely refer to the statement. Some very old references need to be replaced with the recent study stating the fact.

·          The methodology needs to be revised in a more organized way by adding a sub-heading for each step for clear understanding.

·         The picture/ illustration of the methodology should be expressed in a single composite image/ illustration following the sequence.

·         The statistical analysis should mention in the last paragraph of the methodology with the detail of the software and specific test used in this study

·         There is a scope of multiple comparisons, and correlation of data with patient-related factors. Please consult with a statistician to improve this section.

·         Although the authors clearly mention the inclusion and exclusion criteria, the method of calculating the sample size is missing in the methodology.

·         The result section should be separated from the discussion in order to understand the outcome clearly.

·         Although the author mention comparison between groups, the statistical analysis (p-value) and correlation between factors (R-value) is missing in the result section.

·         Please add the PCR result in the form of qualities result (expression band images) and/ or quantitative data.

·         The discussion section is in lacks citations and a logical explanation of the result obtained in this study.

·         The conclusion section needs to be revised with a more clear and summarized outcome of the study.

Author Response

We begin by thanking you for reviewing our article in order to publication  at this special issue edition  by trying  to respond with the most seriousness and goodwill to the pertinent helpful comments, and start in order, as  is presented in the rewieu request report and will be  found in the attached text with track changes markes  by

 1.Revise English English language and style are fine/minor spell check required - done

  2 .The abstract needs to be re-structured by shortening the methodology and adding highlighted results and a clear conclusion/ outcome of the study.-done

  1.  The Authors have cited 3-6 articles for a single statement. The reviewer’s suggestion would be to use 1 or maximum 2 articles that precisely refer to the statement. Some very old references need to be replaced with the recent study stating the fact.-done
  2. The methodology needs to be revised in a more organized way by adding a sub-heading for each step for clear understanding.-done
  3.  The picture/ illustration of the methodology should be expressed in a single composite image/ illustration following the sequence.-done
  4.  The statistical analysis should mention in the last paragraph of the methodology with the detail of the software and specific test used in this study.- done
  5. There is a scope of multiple comparisons, and correlation of data with patient-related factors. Please consult with a statistician to improve this section.-done
  6. The method of calculating the sample size is missing in the methodology. -done
  7. The result section should be separated from the discussion in order to understand the outcome clearly.-done
  8. Although the author mention comparison between groups, the statistical analysis (p-value) and correlation between factors (R-value) is missing in the result section. –done

11.Please add the PCR result in the form of qualities result (expression band images) and/ or quantitative data.-done

12.The discussion section is in lacks citations and a logical explanation of the result obtained in this study.-done

13.The conclusion section needs to be revised with a more clear and summarized outcome of the study.-done

Thank you .

Reviewer 2 Report

Dear Authors,

This study aims to quantify the main bacteria involved in periodontal pathology and associate them with the other factors involved in the onset of periodontal disease so that an accurate diagnosis can be developed, with profound implications for the therapeutic plan.

The study was in line with the aims of the journal. 

However, the manuscript should be improved.

Introduction

-       Lines 58: Please modify “[1,2,3,4]” to “[1-4]”. Please apply to all the text.

-       Line 96: “)”. Please correct.

-       In my opinion the Introduction Section is too long. Please try to reduce it. In my opinion you should add a brief section in which is reported the link among periodontal and systemic diseases. Pleass discuss and cite: “Periodontal Disease and Vitamin D Deficiency in Pregnant Women: Which Correlation with Preterm and Low-Weight Birth? J Clin Med. 2021 Oct 2;10(19):4578. doi: 10.3390/jcm10194578.” ; “Periodontitis and cardiovascular diseases: Consensus report. J Clin Periodontol. 2020 Mar;47(3):268-288. doi: 10.1111/jcpe.13189.” ; “Periodontal disease and cancer: Epidemiologic studies and possible mechanisms. Periodontol 2000. 2020 Jun;83(1):213-233. doi: 10.1111/prd.12329.” ; “Functional status and oral health in patients with amyotrophic lateral sclerosis: A cross-sectional study. NeuroRehabilitation. 2021;48(1):49-57. doi: 10.3233/NRE-201537.”

Materials and Methods

-       Lines 142-143: “on a group of  250 women and 140 men, subjects aged between 20-70 years”. Please put this information in the Result Section. 

-       Lines 145-146: “of which 35 subjects following periodontal therapy in context of oral complex rehabilitation treatments were selected”. Please put this information in the Result Section. 

-       Line 151: “each group consisting of 125 women and 70 men”. Please put this information in the Result Section. 

-       Line 155-157: Please report this information in a separate section name “Statistical Analysis” and better clarify the section.

-       Lines 168-182: Please report this information in a separate section.

-       Lineas 184-198: Please report this information in a separate section.

The Result and Discussion Sections should be separated

Please follow the Instruction for Authors. https://www.mdpi.com/journal/applsci/instructions.

-       Line 207-2013. The sentence is not clear and too long.

-       A revision of the English language is mandatory. 

References 

-       Please rewrite all the references according to the Instruction for Authors. Journal Articles:
1. AUTHOR 1, A.B.; AUTHOR 2, C.D. Title of the article. ABBREVIATED JOURNAL NAME YearVolume, page range. (
https://www.mdpi.com/journal/applsci/instructions).

Author Response

Good afternoon , and thank you for agreeing to review our article, we respond promptly to your sugestions and requests as follows:

Extensive editing of English language and style required –done

     Lines 58: Please modify “[1,2,3,4]” to “[1-4]”. Please apply to all the text. -done

-       Line 96: “)”. Please correct.-done

-        Introduction Section is too long. Please try to reduce it. In my opinion you should add a brief section in which is reported the link among periodontal and systemic diseases. Pleass discuss and cite: “Periodontal Disease and Vitamin D Deficiency in Pregnant Women: Which Correlation with Preterm and Low-Weight Birth? J Clin Med. 2021 Oct 2;10(19):4578. doi: 10.3390/jcm10194578.” ; “Periodontitis and cardiovascular diseases: Consensus report. J Clin Periodontol. 2020 Mar;47(3):268-288. doi: 10.1111/jcpe.13189.” ; “Periodontal disease and cancer: Epidemiologic studies and possible mechanisms. Periodontol 2000. 2020 Jun;83(1):213-233. doi: 10.1111/prd.12329.” ; “Functional status and oral health in patients with amyotrophic lateral sclerosis: A cross-sectional study. NeuroRehabilitation. 2021;48(1):49-57. doi: 10.3233/NRE-201537.” -done

Materials and Methods

-       Lines 142-143: “on a group of  250 women and 140 men, subjects aged between 20-70 years”. Please put this information in the Result Section. -done

-       Lines 145-146: “of which 35 subjects following periodontal therapy in context of oral complex rehabilitation treatments were selected”. Please put this information in the Result Section. -done

-       Line 151: “each group consisting of 125 women and 70 men”. Please put this information in the Result Section. -done

-       Line 155-157: Please report this information in a separate section name “Statistical Analysis” and better clarify the section.-done

-       Lines 168-182: Please report this information in a separate section.-done

-       Lineas 184-198: Please report this information in a separate section.-done

The Result and Discussion Sections should be separated  -done

-       Line 207-2013. The sentence is not clear and too long.-done

-       A revision of the English language is mandatory. -done

Reference- Please rewrite all the references according to the Instruction for Authors. Journal Articles:. (https://www.mdpi.com/journal/applsci/instructions). -done

Round 2

Reviewer 1 Report

Thank you for the revision

Reviewer 2 Report

Dear Authors,

in my opinion the manuscript is suitable for publication.